# The Pea Oligosaccharides Could Stimulate the In Vitro Proliferation of Beneficial Bacteria and Enhance Anti-Inflammatory Effects via the NF-κB Pathway

**DOI:** 10.3390/foods13040626

**Published:** 2024-02-19

**Authors:** Yongxia Cheng, Ruoqi Zhao, Mingwu Qiao, Yan Ma, Tiange Li, Ning Li, Yue Shen, Xianqing Huang, Lianjun Song

**Affiliations:** 1College of Food Science and Technology, Henan Agricultural University, Zhengzhou 450002, China; chengyx1130@163.com (Y.C.); ruoqi19980606@163.com (R.Z.); qmw0309@126.com (M.Q.); mayan201509@163.com (Y.M.); litiange@henau.edu.cn (T.L.); lining8028@126.com (N.L.); shenyue1020@163.com (Y.S.); hxq8210@126.com (X.H.); 2Zhengzhou City Key Laboratory for Soybean Refined Processing, Zhengzhou 450002, China

**Keywords:** oligosaccharide, in vitro fermentation, intestinal flora, anti-inflammatory

## Abstract

The oligosaccharides extracted from the seeds of peas, specifically consisting of raffinose, stachyose, and verbascose, fall under the category of raffinose family oligosaccharides (RFOs). The effect of RFOs on intestinal microflora and the anti-inflammatory mechanism were investigated by in vitro fermentation and cell experiments. Firstly, mouse feces were fermented in vitro and different doses of RFOs (0~2%) were added to determine the changes in the representative bacterial community, PH, and short-chain fatty acids in the fermentation solution during the fermentation period. The probiotic index was used to evaluate the probiotic proliferation effect of RFOs and the optimal group was selected for 16S rRNA assay with blank group. Then, the effects of RFOs on the inflammatory response of macrophage RAW264.7 induced by LPS were studied. The activity of cells, the levels of NO, ROS, inflammatory factors, and the expression of NF-κB, p65, and iNOS proteins in related pathways were measured. The results demonstrated that RFOs exerted a stimulatory effect on the proliferation of beneficial bacteria while concurrently inhibiting the growth of harmful bacteria. Moreover, RFOs significantly enhanced the diversity of intestinal flora and reduced the ratio of Firmicutes-to-Bacteroides (F/B). Importantly, it was observed that RFOs effectively suppressed NO and ROS levels, as well as inflammatory cytokine release and expression of NF-κB, p65, and iNOS proteins. These findings highlight the potential of RFOs in promoting intestinal health and ameliorating intestinal inflammation.

## 1. Introduction

The pea oligosaccharides primarily consist of RFOs, which are widely distributed throughout the plant kingdom and represent a distinct class of water-soluble sugars, exhibiting a content second only to sucrose [1]. Furthermore, RFOs are the predominant water-soluble sugars in legumes and have demonstrated promising potential as prebiotics in the food industry [2]. The RFOs refer to sucrose molecules with one or more galactosidyl groups attached at the glucose-C6 position through alpha-1, 6-glucoside bonds. Raffinose (trimer) represents a subset of RFOs, which also includes stachyose (tetramers), verbascose (pentamers), sineosaccharides (hexamers), and as yet unnamed long-chain oligosaccharides. These derivatives of sucrose can extend up to nine sugars in length [3]. The oligosaccharides are believed to exert regulatory effects on the intestinal microbiota and alleviate intestinal inflammation. Feeding rats a diet rich in raffinose significantly enhances the abundance of *Lactobacillus* spp., reduces *C. perfringens*, and markedly elevates the levels of acetic and propionic acids in the caecum [4]. The oral administration of stachyose demonstrated the potential to mitigate dysbiosis of the intestinal flora and inflammatory response induced by dextran sulfate sodium (DSS) [5]. The verbascose derived from mung bean is believed to possess immunomodulatory properties and exhibit potential for ameliorating inflammatory responses in mice [6].

The intestinal microbiota plays a pivotal role in maintaining intestinal homeostasis. By modulating immune responses and promoting the production of short-chain fatty acids, the intestinal microbiota exerts anti-inflammatory effects within the gut [7]. The regulation of intestinal inflammation by oligosaccharides is widely acknowledged to involve the modulation of flora composition. Additionally, oligosaccharides themselves could mitigate proinflammatory cytokines through the regulation of intestinal immune response [8]. Inflammation is an energetic tissue defense response to damaging stimuli and an attempt by the organism to rid itself of the stimulus and then initiate the tissue healing process [9]. Lipopolysaccharides (LPS), which are endotoxins derived from the outer membrane of Gram-negative bacteria, could recruit a cohort of inflammatory mediators such as macrophages to inflammatory sites and initiate intracellular cascades [10,11]. Macrophages play a crucial role in the immune response of organisms by excreting a series of proinflammatory cytokines, nitric oxide (NO), signaling proteins, and other inflammatory mediators and are activated by stimuli such as proinflammatory cytokines, including tumor necrosis factor-α (TNF-α) and interleukin 6 (IL-6) [12]. NO is regulated by inducible NO synthase (iNOS) and reacts with peroxides to boost inflammatory processes [13]. The anti-inflammatory effects of oligosaccharides may be attributed to their ability to inhibit the production of iNOS and inflammatory factors, as well as other signaling pathways.

Pea oligosaccharides, extracted from peas, consist primarily of raffinose, stachyose, and verbascose. Despite the literature suggesting the potential of these oligosaccharides in regulating intestinal flora and preventing chronic intestinal inflammation, no relevant reports for pea oligosaccharides have been published on this topic to date. Therefore, this study aimed to extract and purify pea oligosaccharides for evaluating their regulatory effects on intestinal microorganisms through simulated in vitro fermentation. Additionally, cell experiments were conducted to explore the mechanism of immunomodulation. These findings aim to provide insights for the development and application of pea oligosaccharides in healthy functional foods.

## 2. Materials and Methods

### 2.1. Pea Oligosaccharides and Other Chemicals

Pea oligosaccharides were extracted from pea powder by ultrasonic-assisted water extraction under the conditions of 1:12 g/mL solid–liquid ratio, 30 s under 700 W microwave, 22 min under 190 W ultrasonic, and being decolorized by AB-8 macroporous resin; the components of the pea oligosaccharides are listed in Appendix A.

Brain Heart Infusion Broth (BHI), Lactobacilli MRS agar, Bifidobacterium agar medium, Clostridium perfringens agar medium, and eosin-methylene blue agar were purchased from Haibor Biology (Qingdao, China). Lactic acid, acetic acid, propionic acid, butyric acid standard, DMEM high-glucose medium, trypsin, PBS, LPS, and loaded buffer Rainbow 245 broad-spectrum protein products were purchased from Soleibao (Beijing, China). RAW264.7 macrophages were purchased from Henan Academy of Agricultural Sciences (Zhengzhou, China). SDS-PAGE electrophoresis solution, BCA protein concentration determination box, and ROS kit were purchased from Biyuntian (Shanghai, China). MTT reagent was purchased from Punosei Bio (Wuhan, China). Fetal bovine serum (FBS) and DMSO were purchased from Gibco (Grand Island, NY, USA). TNF-α and IL-6 inflammatory cytokine kit were purchased from Shanghai Enzyme Link (Shanghai, China). NF-κB p65/p-p65 and iNOS antibody were purchased from Sanying (Wuhan, China).

### 2.2. In Vitro Fermentation Experiment

#### 2.2.1. Enumeration of Bacteria

According to the method of AI, which has been modified, normal mouse feces were collected in a sterile stool collection box and immediately transferred to an anaerobic operating box with sterile phosphate buffer (PBS, 0.1 mol/L) [14]. Mouse feces were prepared with PBS buffer at 1:10 (*w*/*v*) ratio. After being thoroughly mixed by vortex oscillator, fecal homogenate was obtained by mixing and filtering with 3 layers of sterile gauze for inoculation. The fecal bacteria suspension prepared in advance was added into the basic medium with 10% bacterial inoculum quantity, each bottle was packed into 20 mL medium, RFOs were added at 0%, 0.5%, 1%, 1.5%, and 2%, and glucose was supplemented to 2%, respectively. In vitro fermentation was carried out in a sterile triangle flask and incubated at 37 °C in an anaerobic box. The samples were measured at 0, 6, 12, 24, and 48 h of fermentation.

#### 2.2.2. Determination of PH Value of Fermentation Solution

After collecting fermentation liquid at each time point, it was quickly immersed in ice water to stop fermentation. The pH value of the supernatant was then measured with a precision pH meter.

#### 2.2.3. Prebiotic Index

The prebiotic index (*PI*) was defined as the growth relationship of each bacterium during fermentation [15], as determined by Equation (1):(1)$$PI=\frac{Bif-Esc}{Total}+\frac{Lac-Clos}{Total}$$
where *Bif*, *Esc*, *Lac*, and *Clos*, respectively, denote the ratio of the quantity of *Bifidobacterium*, *Escherichia coli*, *Lactobacillus*, and *Clostridium perfringens* in fermentation broth at different time points to that of the corresponding bacteria at 0 h; *Total* represents the ratio of total bacteria quantity in fermentation broth at different time periods to total intestinal bacteria quantity in fermentation broth at 0 h.

#### 2.2.4. Determination of SCFA Content in Fermentation Broth

A total of 0.2 mL of metaphosphoric acid (25% *v*/*v*) was added to a 2 mL centrifuge tube and gently swirled for 2 min. After thorough mixing, the mixture was centrifuged at a speed of 10,000 rpm for 15 min. Following centrifugation, the supernatant was collected and stored at −20 °C after passing through a 0.22 μm organic phase filtration membrane. Ion chromatographic analysis was performed using an ICS-5000+ ion chromatograph from Thermo Field, Franklin, TN, USA, with an IonPac AS11 column (300 × 4.0 mm). The eluent used was NaOH and the sample size injected was 10 μL. Detection was carried out using an inhibitory conductivity detector from Thermo Field, Franklin, TN, USA, with a flow rate of 1 mL/min at room temperature.

#### 2.2.5. Sequencing of 16S rRNA Gene of Intestinal Flora

Based on the results of in vitro fermentation, 16S rRNA gene sequencing was conducted on both the control group and the group supplemented with 1.5% RFOs. Fecal samples were collected and stored at −80 °C for 24 h before being sent to Shanghai Meiji Biological Pharmaceutical Co., Ltd. for microbial community diversity analysis. The bacterial DNA was extracted, and primers 338F and 806R were selected for amplification using the Miseq 2 × 300 platform.

### 2.3. Cell Experiment

#### 2.3.1. Cell Culture

DMEM medium supplemented with 10% fetal bovine serum (containing L-glutamine, without sodium pyruvate) was prepared. The cell suspension was then distributed into T25 cell culture bottles and 4–5 mL of medium per bottle was gently added. After gentle shaking, the cells were cultured in a 37 °C incubator with 5% CO_2_. Daily observations were made to monitor the cellular status.

#### 2.3.2. Cell Viability Assay

The MTT method was employed to assess the impact of RFOs on cell viability. Briefly, logarithmically growing cells were seeded into 96-well plates at a density of 7 × 10^4^ cells/well and incubated at 37 °C for 24 h. Subsequently, the cells were washed with PBS and divided into three groups: blank group, LPS group (1 μg/mL), and LPS + RFOs co-culture group for an additional 24 h. After incubation, the supernatant was aspirated and replaced with 100 μL of MTT solution (0.5 mg/mL) in each well. Following a further 4 h of culture, the supernatant was removed and replaced with 100 μL of DMSO. The absorbance at 570 nm was measured using an enzyme label [16].

#### 2.3.3. NO Content Determination

NO release was quantified using the Griess method [17]. Cells in logarithmic growth phase were seeded at a density of 7 × 10^4^ cells/well in 24-well plates with a volume of 500 μL per well. The cells were incubated at 37 °C and treated with blank, LPS, or LPS + RFOs groups for 24 h. Subsequently, 100 μL supernatant from each well was transferred to a new 96-well plate, followed by the addition of 100 μL Griess reagent. After incubation at room temperature for 20 min, the absorbance at 540 nm was measured.

#### 2.3.4. ROS Content Determination

The detection was performed using the ROS kit, while referring to and modifying the method proposed by Wook et al. [18]. The cells in logarithmic growth phase were harvested and prepared as a cell suspension with an appropriate concentration. After determining the cell density through counting, the cells were seeded into 24-well plates at a concentration of 2 × 10^5^ cells/well, with 500 μL of cell suspension per well, ensuring uniform distribution at the bottom of each well. Once the cells reached 70–80% confluency, they were treated with different concentrations (blank, LPS (1 μg/mL), LPS + 50, LPS + 100, and LPS + 150 μg/mL) of RFOs for 24 h according to experimental groups. Following treatment, the cell supernatant was discarded and washed three times with serum-free medium. Subsequently, H2DCFDA fluorescent probe was added to serum-free medium (diluted at a ratio of 1:1000) and incubated at 37 °C for 30 min. After washing away the ROS probe three times using serum-free medium, fluorescence was observed under a fluorescence microscope and images were captured. The fluorescence spectrophotometer was configured with an excitation wavelength of 488 nm and an emission wavelength of 525 nm.

#### 2.3.5. TNF-α and IL-6 Inflammatory Factors

The proinflammatory factors secreted by RAW264.7 cells were quantified using enzyme-linked immunosorbent assay (ELISA) from Shanghai Enzyme Link (Shanghai, China), and the cells were categorized based on Maria et al.’s classification [19].

#### 2.3.6. Immunofluorescence Assay

The expression of NF-κBp65/p-p65 and iNOS proteins was detected by immunofluorescence method. The cells were cultured in groups according to the method of Hyeon et al. [20].

### 2.4. Statistical Analysis

Statistical analyses were performed using SPSS statistical software (version 19.0, SPSS Inc., Chicago, IL, USA). The significance of treatment effects was evaluated using analysis of variance (ANOVA) and Tukey’s test. Data are reported as mean ± SE.

## 3. Results

### 3.1. The pH Value of Each Group Changed during Fermentation

The effects of different dosages of RFOs on pH changes during fermentation are shown in Table 1. It can be observed that the pH value of the control group initially decreased and then increased during the fermentation process. This phenomenon could be attributed to the early production of acid by lactic acid bacteria, followed by an increase in solution pH due to protein utilization by *E. coli* and other bacteria in later stages. Both the glucose group and the 0.5–2% RFOs group exhibited a gradual decline in pH value, leading to a shift from neutral to acidic conditions throughout fermentation. Notably, at the end of fermentation, the glucose group displayed the lowest pH value, possibly due to anaerobic fermentation generating significant amounts of lactic acid, while RFOs promoted intestinal acidity.

### 3.2. Changes in Representative Flora during Fermentation

In this study, *Lactobacillus*, *Bifidobacterium*, *Clostridium perfringens*, and *Escherichia coli* were selected as test organisms to investigate the impact of RFOs on the microbial flora composition during simulated intestinal fermentation. The findings are presented in Figure 1. The population of *Lactobacillus* in each group exhibited an initial increase followed by a subsequent decrease during the 0–48 h fermentation period (Figure 1A). Compared to the control group, the addition of 0.5% RFOs significantly enhanced *Lactobacillus* growth from 24 h to 48 h of fermentation. Similarly, supplementation with 1% RFOs notably promoted *Lactobacillus* proliferation from 12 h to 48 h of fermentation, while the inclusion of 1.5% RFOs specifically stimulated *Lactobacillus* growth at the 24 h mark. Notably, among these groups, the supplementation with 1% RFOs demonstrated the highest number of *Lactobacillus* at the end of a 24 h fermentation period, reaching a count of approximately 2.49 × 10^9^ cfu/mL. The results demonstrated a significant increase of approximately 90.07% and 54.65%, respectively, compared to both the control and glucose groups at equivalent time points, indicating that optimal levels of RFOs effectively promote *Lactobacillus* proliferation.

The total number of *Bifidobacterium* colonies exhibited changes as depicted in Figure 1B. Over the course of 0–48 h fermentation, the population of *Bifidobacterium* gradually increased and subsequently reached a stable level in each group. Notably, at the 12th hour of fermentation, the 1.5% RFOs group displayed a significant increase in *Bifidobacterium* count, which was, respectively, 44.78% and 42.76% higher than that observed in the blank group and glucose group at the same time point. However, no significant differences were observed between groups regarding *Bifidobacterium* count at both 24 h and 48 h time points.

The total number of *Clostridium perfringens* colonies exhibited changes, as depicted in Figure 1C. During the fermentation period of 0–48 h, both the blank group and glucose group showed an increase in *Clostridium perfringens* count. However, the 0.5% to 2% RFOs group demonstrated a significant decrease in *Clostridium perfringens* count compared to the blank group and glucose group after 12 h of fermentation.

As depicted in Figure 1D, the population of *Escherichia coli* exhibited an upward trend during fermentation in both the blank and glucose groups. Specifically, the number of *Escherichia coli* in the blank group ranged from 1.23 × 10^5^ to 1.44 × 10^9^ cfu/mL, while, in the glucose group, it ranged from 6.73 × 10^4^ to 5.70 × 10^8^ cfu/mL. Subsequently, upon the addition of varying amounts of RFOs, a significant reduction was observed in the number of *Escherichia coli*, which inversely correlated with increasing levels of RFOs.

### 3.3. The Changes in Prebiotic Index and Short-Chain Fatty Acid in Each Group during Fermentation

The prebiotic index of RFOs is illustrated in Figure 2A. The 0.5% RFOs group exhibited a beneficial effect after 48 h of fermentation, while the 1% RFOs group showed a probiotic effect at 24 h of fermentation. Both the 1.5% and 2% RFOs groups demonstrated a beneficial effect from 24 to 48 h of fermentation, with the probiotic effect being most pronounced during this period. When different amounts of RFOs were added for a duration of 24 h, the probiotic index in each group ranked as follows: 1.5% RFOs > 2% RFOs > 1% RFOs > 0.5% RFOs (from highest to lowest). This suggests that an addition of approximately 1.5% RFOs yields optimal results, with higher quantities leading to improved effects. The anaerobic fermentation process, as depicted in Figure 2B, results in a substantial production of lactic acid, except for the blank group. Although lactic acid is not classified as a short-chain fatty acid, it does play a crucial role in pH regulation. Notably, the inclusion of 1.5% and 2% RFOs significantly enhances acetic acid production by 1.85-fold and 2.33-fold compared to the glucose group. The samples from the 1.5% RFOs group, which underwent a 24 h fermentation based on the aforementioned results, were selected for subsequent analysis using 16S rRNA sequencing.

### 3.4. Effects of RFOs on Microbial Diversity

In the analysis of intestinal flora, Alpha diversity serves as a metric for assessing the richness and evenness of microbial communities within an individual’s gut. The investigation into microbial diversity in community ecology employs diversity analysis to capture both the abundance and variety of microbial populations. This entails utilizing several statistical indices to estimate species abundance and environmental community diversity. The Chao1 index and ACE index are employed as measures of community richness, with higher values indicating greater richness. Conversely, the Shannon index and Simpson index are utilized as indicators of community diversity, with higher Shannon values reflecting increased diversity, while lower Simpson index values indicate higher levels of community diversity [21]. Firstly, the impact of RFOs on the Alpha diversity of intestinal microbes was assessed using Chao1 index, ACE index, Shannon index, and Simpson index. As depicted in Figure 3A,B, both the Chao1 index and ACE index of the sample group were significantly higher than those of the BLK group, indicating an enhanced richness of intestinal flora in the sample group. However, no significant differences were observed in terms of Shannon index (Figure 3C) and Simpson index (Figure 3D) between the two groups, suggesting that RFOs did not influence bacterial diversity.

The beta diversity of intestinal microorganisms serves as an indicator for community composition across different groups. Principal component analysis (PCA) and principal co-ordinates analysis (PCoA) are both methods used to analyze beta diversity. PCA and PcoA are commonly employed to determine the similarity in composition of intestinal microbial communities. In PCA, the first and second principal components represent the reduction in variance observed within each sample, while the distance between different community samples reflects their dissimilarity. Similarly, PcoA also utilizes the first and second principal components as axes, which are determined based on a dissimilarity matrix. Consequently, greater distances between community samples in these analyses indicate larger differences in microbial community structure [22]. The results of the species-level principal component analysis are presented in Figure 3E. As depicted, the first principal component contributed 48.43% to the test results, while the second principal component contributed 12.01%, resulting in a cumulative contribution exceeding 50%. Notably, the blank group and the 1.5% RFOs group exhibited distinct clustering with improved separation on the figure, providing compelling evidence for significant differences in intestinal flora composition compared to the blank group. The PcoA analysis employed Weighted Unifrac distance and is illustrated in Figure 3F. The first and second principal components accounted for contributions of 96.14% and 2.13%, respectively. Consistently, both groups were distributed across different regions on the co-ordinate map, further confirming substantial dissimilarities in intestinal flora composition between them. Importantly, RFOs demonstrated an effective ability to modulate intestinal flora structure and enhance community stability.

### 3.5. Effect of RFOs on Intestinal Flora Composition

The effect of adding 1.5% RFOs on the intestinal flora at the phylum level is illustrated in Figure 4A. At this level, the microorganisms present in the fermentation solution primarily consist of Firmicutes, Bacteroidetes, Proteobacteria, and Actinobacteria. Among these, Bacteroides and Firmicutes are dominant intestinal flora, accounting for over 95% of total intestinal microbes in rats. In comparison to the control group, the 1.5% RFOs group exhibited a significant increase in the relative abundance of Firmicutes and Bacteroidetes, while Actinobacteria and Proteobacteria demonstrated a notable decrease. The Firmicutes-to-Bacteroides (F/B) value for the 1.5% RFOs group was 141.2, which was markedly lower than the control group’s value of 442.36. The composition of intestinal microbes at the genus level is illustrated in Figure 4B. In the 1.5% RFOs group, there was a notable increase in Lactobacillus along with Enterococcus, Romboutsia, Clostridium-sensu-stricto-1, unclassified-o-Lactobacillales, and Turicibacter. Conversely, Bifidobacterium, Weissella, and Escherichia-Shigella exhibited a significant decrease. Notably, the control group had an abundance of Escherichia-Shigella at approximately 2.89% ± 0.64%, mentioning that almost no *Escherichia* was detected within the 1.5% RFOs group.

### 3.6. Anti-Inflammatory Effect of RFOs

#### 3.6.1. Effects of RFOs on the Levels of NO and ROS in Model Cells

The MTT method was employed to assess the impact of RFOs on macrophage viability in order to determine the appropriate sample concentration, as depicted in Figure 5A. The cell proliferation rate measured in the control group was 100%, while, after treatment with RFOs samples at concentrations of 25, 50, 100, 150, and 200 μg/mL, the cell proliferation rates were found to be 107.94%, 107.45%, 106.58%, 97.10%, and 53.84%, respectively. When the concentration of RFOs was below or equal to 150 μg/mL, no significant effect on macrophage activity was observed. However, when the concentration reached up to 200 μg/mL, a notable decrease in cell proliferation rate occurred, indicating low cytotoxicity of RFOs within the range of 25~150 μg/mL. In this study, RFOs at concentrations of 50, 100, and 150 μg/mL were selected for investigating their anti-inflammatory effects.

The macrophage RAW264.7 was stimulated with LPS to establish an inflammation model, and the levels of NO and reactive oxygen species (ROS) were measured to assess the anti-inflammatory effects of RFOs, as depicted in Figure 5B,C. As illustrated in Figure 5B, there was a significant increase in NO production in the LPS (1 μg/mL) group compared to the control group, confirming successful induction of the inflammation model. NO serves as a crucial mediator in various biological functions, and reduction in its release can be indicative of improved inflammatory response within cellular models [23]. In the experimental groups of RFOs in this study, as the concentration of RFOs increased, there was a gradual decrease in the release of NO. The corresponding inhibition rates were 17.54%, 36.98%, and 62.89%, respectively, indicating that RFOs could significantly reduce NO secretion in LPS-induced inflammatory model cells in a concentration-dependent manner.

The intracellular levels of reactive ROS were assessed using fluorescent probes, enabling the analysis of cellular oxidative stress levels [24]. The intracellular ROS levels were approximately three times higher after LPS treatment compared to the blank group, as depicted in Figure 5C. Within the concentration range of 50 to 150 μg/mL, intervention with RFOs significantly attenuated ROS levels. Figure 5D displays fluorescence microscope images of cells from each experimental group, and the corresponding change in fluorescence intensity further supports these findings. These results suggest that, while RFOs cannot completely eliminate LPS-induced ROS, they do exert a certain protective effect on cells.

#### 3.6.2. Effects of RFOs on the Levels of TNF-α and IL-6 in Model Cells

Figure 6A,B depict the impact of various concentrations of RFOs treatment on proinflammatory factor levels in macrophages during the LPS-induced inflammatory response. Compared to the control group, there was a significant increase observed in the release of TNF-α and IL-6 in the LPS group. However, as the concentrations of RFOs intervention increased, there was a subsequent decrease observed in TNF-α and IL-6 levels.

#### 3.6.3. Effect of RFOs on Expression of NF-κB p-p65/p65 and iNOS Proteins in Model Cells

In this study, we evaluated protein levels of NF-κB p-p65/p65 and iNOS in macrophages using Western blot analysis (Figure 7). As depicted in Figure 7A, there was a significantly higher level of p65 protein phosphorylation after LPS stimulation for 24 h compared to the control group. Treatment with RFOs dose-dependently inhibited LPS-induced upregulation of p-p65/p65 expression. Furthermore, Figure 7B demonstrates that iNOS protein levels were significantly increased after LPS stimulation for 24 h compared to the control group. Once again, treatment with RFOs dose-dependently suppressed LPS-induced iNOS expression. These findings suggest that RFOs possess potent anti-inflammatory effects and may regulate the body’s inflammatory response through modulation of the *NF-κB* pathway.

## 4. Discussion

This study aimed to investigate the impact of pea oligosaccharides on microflora composition during in vitro fermentation of mouse feces, as well as their potential role in mitigating LPS-induced inflammation in RAW264.7 cells.

Gut microbes play a crucial role in maintaining homeostasis and can be categorized into symbiotic bacteria, opportunistic pathogens, and pathogenic bacteria based on their interactions with the host [25]. Symbiotic bacteria primarily consist of obligate anaerobes such as *Lactobacillus*, *Bifidobacterium*, *Bacteroides*, etc., which metabolize beneficial substances and contribute to overall health. Opportunistic pathogens are mostly facultative anaerobic bacteria like *Escherichia coli*, *Enterococcus*, and *Enterobacter* that proliferate when there is disruption in intestinal metabolism. Pathogenic bacteria, such as *Pathogenic Salmonella* and *Clostridium perfringens*, are nonresident intestinal flora that elicit diseases upon human ingestion. The intestinal microenvironment is a complex ecosystem and reaction system. In the growth and metabolic activities of microorganisms, the pH in the intestine plays a crucial role in regulating enzyme activity, making it an important parameter for assessing intestinal health [26]. Research has demonstrated that a decrease in intestinal pH promotes the proliferation of beneficial microorganisms, while an increase in pH facilitates the growth of certain harmful bacteria or opportunistic pathogens [27].

The impact of pea oligosaccharides on the intestinal microbiota was quantitatively evaluated in this study using the prebiotic index, which represents the correlation between specific bacteria and the total bacterial count in mouse intestinal flora fermentation broth [15]. *Lactobacillus* and *Bifidobacterium* are the predominant probiotics in the intestinal flora [28], while *Escherichia coli* represents opportunistic pathogens [29] and *Clostridium perfringens* represents pathogenic bacteria [30]. These four bacterial species are commonly found in the gut and can serve as indicators of gut health to a certain extent; hence, they were selected for this study. *Bifidobacterium* and *Lactobacillus* demonstrate positive values, while *Clostridium perfringens* and *Escherichia coli* exhibit negative values [31].

The results presented above demonstrate that RFOS has the ability to lower the pH value during fermentation and enhance the proliferation of *Lactobacillus* and *Bifidobacterium*, while inhibiting the growth of *Clostridium perfringens* and *Escherichia coli*. Consequently, RFOs possess an improved prebiotic index. *Lactobacillus* and *Bifidobacterium* play a crucial role in maintaining the balance of intestinal flora by producing organic acids that lower intestinal pH, inhibiting the proliferation of harmful bacteria [32]. The observed decrease in *Clostridium perfringens* in the RFOs group may be attributed to the increased abundance of *Lactobacillus* and *Bifidobacterium*. *Escherichia coli*, a member of the Escherichia genus, was found to be significantly inhibited in its proliferation during in vitro fermentation by RFOs. Furthermore, the reduction in *Escherichia coli* levels suggests a healthy gut environment. The presence of Actinomycetes is associated with a significant number of saprophytic organisms, which can contribute to the development of tuberculosis, abscesses, and leprosy [33]. Similarly, an increase in Proteobacteria microorganisms may disrupt the balance of intestinal flora, leading to low-grade inflammation and potentially chronic colitis [34]. In this study, it was observed that the relative abundance of Actinobacteria and Proteobacteria in the 1.5% RFOs group was significantly reduced, providing further evidence for the inhibitory effect of RFOs on the growth of intestinal pathogens. In conclusion, RFOs exhibit potential for modulating intestinal flora composition and promoting intestinal health through their ability to enhance probiotic populations while suppressing conditioned and pathogenic bacteria growth.

*Lactobacillus*, a typical genus of Firmicutes, plays a crucial role as beneficial flora in the gut. The significant increase in *Lactobacillus* may be the primary reason for the rise in Firmicutes abundance. Acidic conditions favor the growth of Firmicutes [35], and pH reduction during in vitro fermentation can also promote their proliferation. Bacteroidetes primarily inhabit the distal intestine, where they ferment nondigestible polysaccharides to generate short-chain fatty acids that serve as an energy source for the body [36]. Dietary fiber and polysaccharides have been reported to impede weight gain by promoting a higher relative abundance of Bacteroidetes [37]. The composition of Firmicutes and Bacteroides in gut microbiota has been identified as potential biomarkers indicating susceptibility to obesity, with obese individuals exhibiting significantly elevated ratios of Firmicutes-to-Bacteroides (F/B) compared to healthy individuals [38]. In this study, the F/B value for the 1.5% RFOs group was markedly lower than the control group, suggesting that RFOs possess potential in modulating intestinal flora to prevent obesity.

The primary byproducts of carbohydrate fermentation in the gut microbiota are short-chain fatty acids, which play a crucial role in regulating lipid and sugar metabolism, promoting immune homeostasis, treating inflammatory diseases, and contributing to human health [39]. The acetic acid level in the 1.5% RFOs group was significantly increased, which may be due to the increase in *Lactobacillus* and *Bifidobacterium* abundance, which aligns with the findings reported by Hui Cao et al. [40].

LPS could activate the innate immune response of macrophages and induce the production of a significant amount of inflammatory cytokines, including tumor necrosis factor-α (TNF-α) and interleukin-6 (IL-6), which are widely recognized as excellent biomarkers for inflammation [41]. Among these markers, TNF-α plays a pivotal role in the pathogenesis of various diseases, particularly chronic inflammation that links rheumatoid arthritis, atherosclerosis, and impaired insulin sensitivity. Prolonged elevation of TNF-α level could inflict significant harm on the body [42]. The IL-6 cytokine plays a pivotal role in the modulation of immune responses and is implicated in the pathogenesis of various diseases, including autoimmune disorders, chronic inflammatory conditions, and cancer [43]. The inhibition by RFOs on TNF-α and IL-6 release in model cells suggests its potent anti-inflammatory effect and potential for preventing chronic diseases. The NF-κB pathway is a critical signaling pathway involved in inflammation and has been shown to participate in the transcription and regulation of various inflammatory mediators [44]. As a significant member of the NF-κB protein family, the p65 subunit undergoes phosphorylation to initiate gene transcription, synthesize inflammatory factors, and further induce and exacerbate inflammatory damage [45]. The iNOS is an enzyme regulated by the NF-κB pathway that plays a pivotal role in synthesizing proinflammatory mediator NO, producing substantial amounts of NO over an extended period [45,46]. Studies have indicated that bacterial-invasion-induced intestinal inflammation could promote iNOS expression [45], highlighting its significance in the inflammatory response. The RFOs inhibit P-P65/p65 protein expression as well as iNOS protein expression (*p* < 0.05), suggesting that their anti-inflammatory effect on LPS-induced macrophage RAW264.7 cells may be mediated through NF-κB signaling pathway activation.

## 5. Conclusions

The results demonstrated that RFOs exhibited a stimulatory effect on the proliferation of probiotics, while inhibiting the growth of opportunistic pathogens and pathogenic bacteria during mouse fecal fermentation. Additionally, RFOs effectively modulated the F/B ratio by enhancing the relative abundance of Firmicutes and Bacteroidetes, thereby promoting microbial diversity and facilitating acetic acid production. Importantly, cell-based assays revealed no cytotoxicity of RFOs towards macrophage RAW264.7 cells; instead, they displayed supportive effects on cell growth at concentrations ≤150 µg/mL. Furthermore, it was observed that RFOs have potential to mitigate inflammation through activation of the NF-κB signaling pathway. The in vivo verification of this study is still pending, and further investigations are required in this field. Overall, these findings underscore the potential value of RFOs in regulating intestinal health due to their ability to promote probiotic proliferation and possess potent anti-inflammatory properties.

## Figures and Tables

**Figure 1 foods-13-00626-f001:**
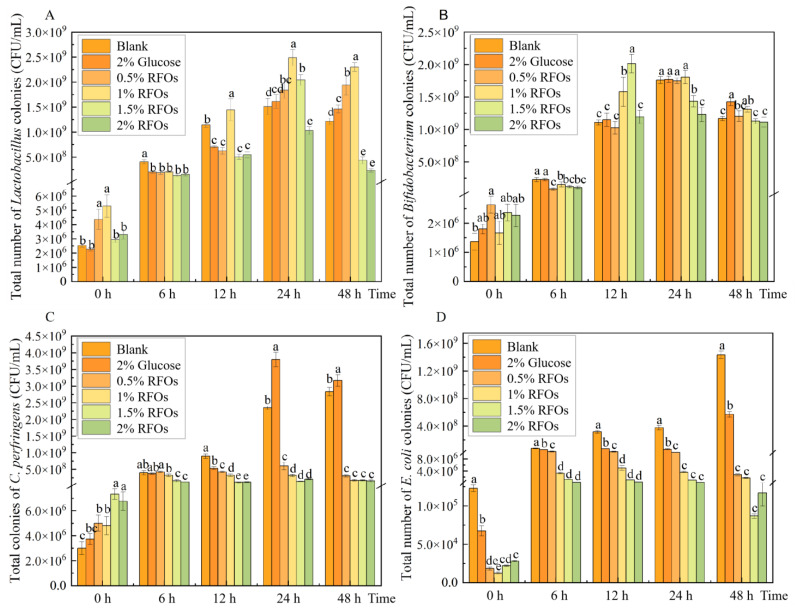
Changes in representative flora during fermentation. (**A**) *Lactobacillus*; (**B**) *Bifidobacterium*; (**C**) *Clostridium perfringens*; (**D**) *Escherichia coli*. Values with different letters at the same time represent significant differences (*p* < 0.05).

**Figure 2 foods-13-00626-f002:**
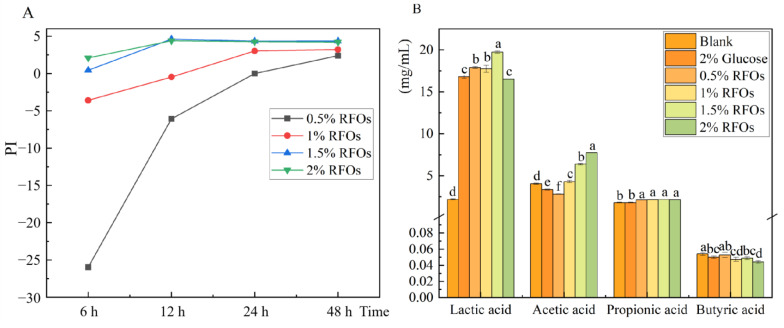
The changes in prebiotic index (PI) and short chain fatty acid in each group during fermentation. (**A**) Change in prebiotic index at 6 h, 12 h, 24 h, and 48 h fermentation; (**B**) changes in lactic acid, acetic acid, propionic acid, and butyric acid during fermentation. Values with different letters in the same index represent significant differences (*p* < 0.05).

**Figure 3 foods-13-00626-f003:**
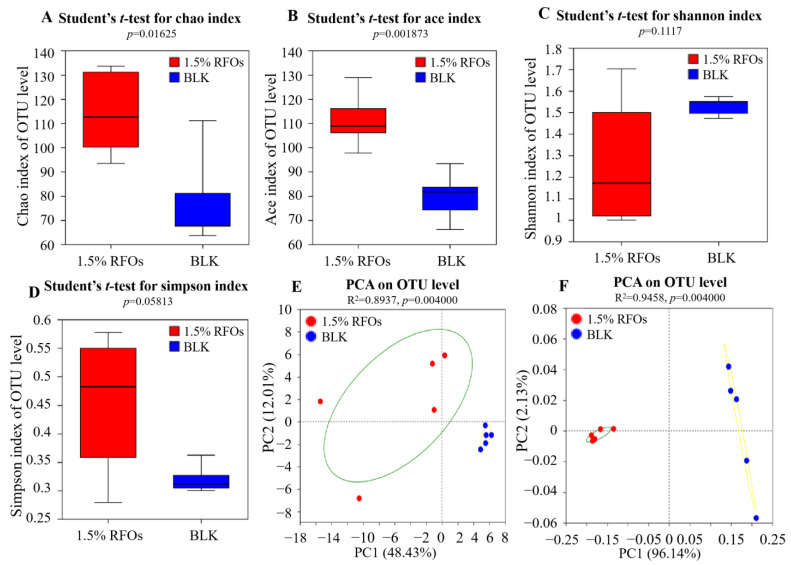
Effects of the RFOs on intestinal flora diversity. (**A**) Chao 1 index; (**B**) ACE index; (**C**) Shannon index; (**D**) Beta diversity; (**E**) PCA; (**F**) PCoA.

**Figure 4 foods-13-00626-f004:**
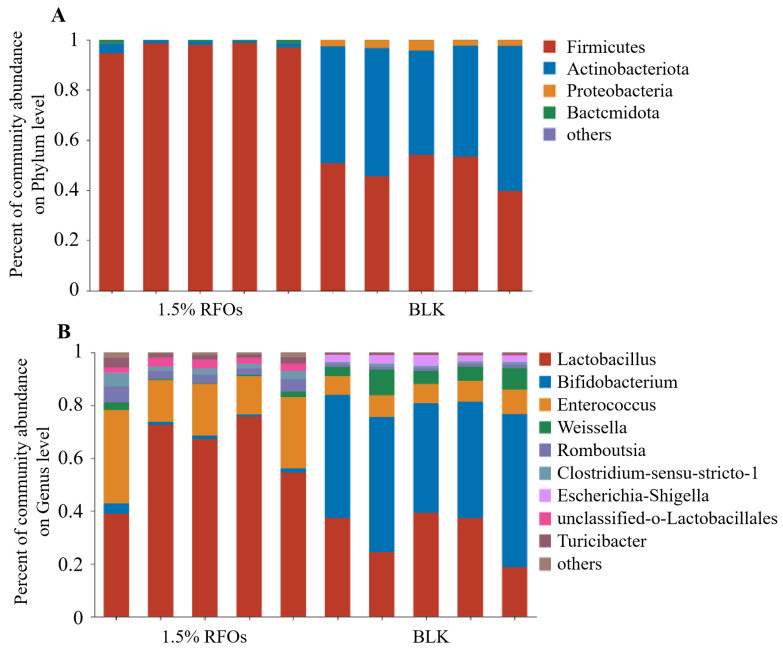
Effects of RFOs on intestinal flora composition. (**A**) Phylum level; (**B**) genus level.

**Figure 5 foods-13-00626-f005:**
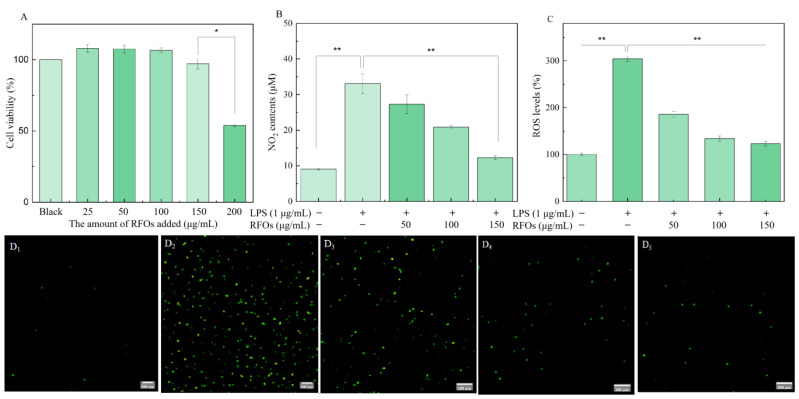
The levels of NO and ROS in model cells. (**A**) Cell viability; (**B**) NO content; (**C**) ROS levels; (**D**) the fluorescence microscope images of (**D1**) control group, (**D2**) LPS (1 μg/mL) group, (**D3**) LPS + RFOs (50 μg/mL) group, (**D4**) LPS + RFOs (100 μg/mL) group, and (**D5**) LPS + RFOs (150 μg/mL) group. “*” indicates significant difference at *p* < 0.05 level, “**” means significant difference at *p* < 0.01 level.

**Figure 6 foods-13-00626-f006:**
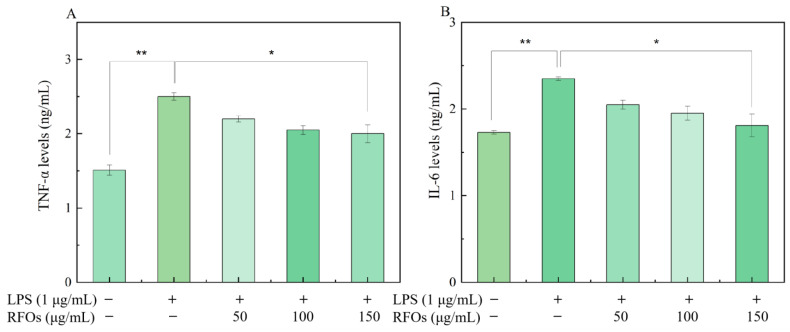
The levels of TNF-α and IL-6 in model cells. (**A**) TNF-α content; (**B**) IL-6 content. “*” indicates significant difference at *p* < 0.05 level; “**” means significant difference at *p* < 0.01 level.

**Figure 7 foods-13-00626-f007:**
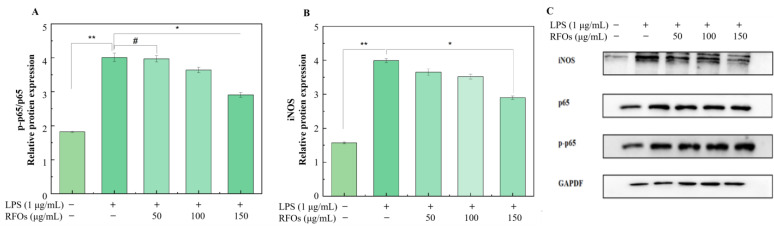
The levels of NF-κB p-p65/p65 and iNOS proteins in model cells. (**A**) NF-κB p65/p-p65 protein; (**B**) iNOS protein; (**C**) inkblot analysis of protein. “#” means the difference is not significant; “*” indicates significant difference at *p* < 0.05 level; “**” means significant difference at *p* < 0.01 level.

**Table 1 foods-13-00626-t001:** The pH values of each group during fermentation.

Groups	Time				
	0 h	6 h	12 h	24 h	48 h
blank	7.34 ± 0.03 ^a^	6.07 ± 0.01 ^e^	6.57 ± 0.02 ^d^	6.90 ± 0.01 ^c^	7.22 ± 0.01 ^b^
2% glucose	7.24 ± 0.02 ^a^	5.05 ± 0.01 ^b^	4.34 ± 0.01 ^c^	3.97 ± 0.02 ^d^	3.76 ± 0.01 ^e^
0.5% RFOs	7.01 ± 0.02 ^a^	5.36 ± 0.02 ^b^	4.72 ± 0.02 ^c^	4.27 ± 0.02 ^d^	3.93 ± 0.01 ^e^
1% RFOs	6.87 ± 0.03 ^a^	5.41 ± 0.04 ^b^	4.75 ± 0.01 ^c^	4.41 ± 0.01 ^d^	4.12 ± 0.01 ^e^
1.5% RFOs	6.66 ± 0.01 ^a^	5.42 ± 0.01 ^b^	4.84 ± 0.01 ^c^	4.49 ± 0.01 ^d^	4.43 ± 0.01 ^e^
2% RFOs	6.52 ± 0.01 ^a^	5.54 ± 0.01 ^b^	5.08 ± 0.01 ^c^	4.84 ± 0.02 ^d^	4.72 ± 0.01 ^e^

The data shown are mean ± standard deviation of three replicates. Values with different letters in the same row are significantly different at *p* < 0.05.

## Data Availability

The data presented in this study are available on request from the corresponding author. The data are not publicly available due to privacy restrictions.

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
