# Peer review of "The Pea Oligosaccharides Could Stimulate the In Vitro Proliferation of Beneficial Bacteria and Enhance Anti-Inflammatory Effects via the NF-κB Pathway"

_foods, 2024, doi:10.3390/foods13040626_

Round 1
Reviewer 1 Report
Comments and Suggestions for Authors
1. The content of pea oligosaccharides needs to be confirmed. Data from the HPLC chromatogram requires standards for each oligosaccharide.
2. Fecal samples should be from humans, not mice.
3. No ethical clearance to obtain feces from mouse or animal experiments.
4. Fecal fermentation in this study does not mimic human colonic fermentation. Experimental design in fecal fermentation is not excellent enough for a high-quality article. It should be in vitro pH-controlled human fecal fermentation.
5. The author makes mistakes in the probiotic index and prebiotic index, as well as 16S rRNA and 16Sr-RNA, Italics of "in vitro", and "in vivo" as well as the scientific name of microorganisms.
6. Prebiotic activity score as defined in this study is not currently reported.
7. SCFA detection by GC with an inhibitory conductivity detector is not used. Normally, it is accepted for GC with an FID detector. Lactic acid should not be detected by GC if no further sample preparation such as derivatization.
Comments on the Quality of English LanguageMinor English correction
Author Response
Editor, Foods
Reviews for: foods-2849523
Title: Effects of Pea Oligosaccharide on Probiotic Proliferation and Modulation of Inflammatory Response
Authors: Yongxia Cheng, Ruoqi Zhao, Mingwu Qiao, Yan Ma, Tiange Li, Ning Li,
Yue Shen, Xianqing Huang, Lianjun Song *
Many thanks for the reviewers’ comments. Below we respond to all of these.
- The content of pea oligosaccharides needs to be confirmed. Data from the HPLC chromatogram requires standards for each oligosaccharide.
Response: The concentrations of sucrose, raffinose, stachyose and verbascose are determined using the curve equation relating peak area to standard product concentration. The standard curve equations of each oligosaccharide have been added to the Supporting Information.
- Fecal samples should be from humans, not mice.
Response: The objective of this study was to assess the impact of pea oligosaccharides on the in vitro fermentation of intestinal bacteria. The use of mice as an alternative to humans for studying the function of oligosaccharides is a common practice[1, 2]. Additionally, conducting in vitro fermentation of mouse feces allows for the assessment of the impact of pea oligosaccharides on intestinal flora. Hence, this study employed mouse feces as a model system.
- No ethical clearance to obtain feces from mouse or animal experiments.
Response: The mice were collected from a sterile environment immediately after their natural defecation. The primary focus of the experiment was on the in vitro cultivation of fecal samples, without causing any harm to the animals during the entire experiment.
- Fecal fermentation in this study does not mimic human colonic fermentation. Experimental design in fecal fermentation is not excellent enough for a high-quality article. It should be in vitro pH-controlled human fecal fermentation.
Response: Thank you for your valuable suggestion. In future similar experiments, we will consider utilizing human fecal matter for in vitro fermentation to enhance the precision of our experimental design.
- The author makes mistakes in the probiotic index and prebiotic index, as well as 16S rRNA and 16Sr-RNA, Italics of "in vitro", and "in vivo" as well as the scientific name of microorganisms.
Response: The complete text has been thoroughly reviewed and subsequently revised in light of these identified errors.
- Prebiotic activity score as defined in this study is not currently reported.
Response: The prebiotic activity score is a quantitative measure utilized for comparing the probiotic effects of dietary prebiotics, assessing the correlation between specific bacteria and the initial bacterial count in the fermentation broth of mice intestinal flora. It is also defined as a prebiotic index in some literatures, and the reference method in this study is the research method of R. Palframan et al[3]. The prebiotic activity score has been uniformly replaced with the prebiotic index throughout the manuscript.
- SCFA detection by GC with an inhibitory conductivity detector is not used. Normally, it is accepted for GC with an FID detector. Lactic acid should not be detected by GC if no further sample preparation such as derivatization.
Response: The analytical technique employed for the quantification of SCFA and lactic acid in this study was ion chromatography. The description in the manuscript was incorrect, and ion chromatography was mistakenly referred to as gas chromatography, which has been rectified.
References:
[1]. Ting, L., L. Xinshan and Y. Xingbin, Stachyose-Enriched α-Galacto-oligosaccharides Regulate Gut Microbiota and Relieve Constipation in Mice[J]. Journal of Agricultural and Food Chemistry 2013. 61(48): 11693-11864.
[2]. Weiwei, C., et al., Effect of Functional Oligosaccharides and Ordinary Dietary Fiber on Intestinal Microbiota Diversity[J]. Frontiers in Microbiology 2017. 8: 01750.
[3]. R. Palframan, G.R. Gibson and R.A. Rastall, Development of a quantitative tool for the comparison of the prebiotic effect of dietary oligosaccharides[J]. Lett Appl Microbiol 2003. 37(4): 281-4.
Reviewer 2 Report
Comments and Suggestions for Authors
The manuscript entitled, ‘Effects of Pea Oligosaccharide on Probiotic Proliferation and Modulation of Inflammatory Response’ explores the extraction and purification of oligosaccharides and investigates their regulatory effects on Lactobacillus, Bifidobacterium, Clostridium perfringens, and Escherichia coli and immunomodulation effects on RAW264.7 macrophages. However, some major concerns should be addressed by the authors prior to any possible consideration of this manuscript to be submitted in the Journal of foods.
1. The #title of the study is very vague and unclear. The authors should think of a specific and informative title.
2. The statement written in the #Abstract, “The oligosaccharides extracted from pea mainly include raffinose family oligosaccharides (RFOs).” is a generalized statement and should be revised by the authors.
3. The #Figures presented in this study are of very poor quality. The authors must prepare suitable figures (e.g., size, shape, aspect ratios, colour, etc.) for scientific presentations.
4. On what basis the authors have chosen Lactobacillus, Bifidobacterium, Clostridium perfringens, and Escherichia coli as their test organisms for studying the regulatory effects of the samples on intestinal flora?
5. What stain did the authors utilize to analyze the ROS status in RAW cells? It must be clarified in the respective section (the camera pixel and scale bar of the microscopic images should also be specified).
6. The authors could have compared the findings of this study (in a separate table) with earlier reported literature (with similar works) to improve the discussion of this manuscript.
7. The important findings and limitations of this should be highlighted in the #Conclusion section.
Comments on the Quality of English LanguageEnglish is fine but the manuscript must be thoroughly checked for confusing and generalized statements and revised accordingly.
Author Response
Editor, Foods
Reviews for: foods-2849523
Title: Effects of Pea Oligosaccharide on Probiotic Proliferation and Modulation of Inflammatory Response
Authors: Yongxia Cheng, Ruoqi Zhao, Mingwu Qiao, Yan Ma, Tiange Li, Ning Li,
Yue Shen, Xianqing Huang, Lianjun Song *
Many thanks for the reviewers’ comments. Below we respond to all of these.
- The #title of the study is very vague and unclear. The authors should think of a specific and informative title.
Response: To enhance the clarity of the study, the paper title has been revised to“The Pea Oligosaccharides Could Stimulate the in vitro Proliferation of Beneficial Bacteria and Enhance anti-inflammatory Effects via the NF-κB pathway.”
- The statement written in the #Abstract, “The oligosaccharides extracted from pea mainly include raffinose family oligosaccharides (RFOs).” is a generalized statement and should be revised by the authors.
Response: The original sentence has been revised to read as follows: "The oligosaccharides extracted from the seeds of peas, specifically consisting of raffinose, stachyose and verbascose, which fall under the category of raffinose family oligosaccharides (RFOs)."
- The #Figures presented in this study are of very poor quality. The authors must prepare suitable figures (e.g., size, shape, aspect ratios, colour, etc.) for scientific presentations.
Response: The images in the manuscript have been modified in terms of size, aspect ratios, clarity, and colour as per the suggested recommendations.
- On what basis the authors have chosen Lactobacillus, Bifidobacterium, Clostridium perfringens, and Escherichia coli as their test organisms for studying the regulatory effects of the samples on intestinal flora?
Response: Lactobacillus and Bifidobacterium are the predominant probiotics in the intestinal flora[1], while Escherichia coli represents opportunistic pathogens[2] and Clostridium perfringens represents pathogenic bacteria[3]. These four bacterial species are commonly found in the gut and can serve as indicators of gut health to a certain extent; hence, they were selected for this study.
- What stain did the authors utilize to analyze the ROS status in RAW cells? It must be clarified in the respective section (the camera pixel and scale bar of the microscopic images should also be specified).
Response: Using H2DCFDA (2, 7-dichlorodihydrofluorescein acetoacetic acid) staining, H2DCFDA itself exhibits no fluorescence and can freely traverse the cellular membrane. Upon intracellular entry, it undergoes hydrolysis by intracellular esterase to generate DCFH; however, DCFH is incapable of permeating the cell membrane, thereby facilitating facile loading of the probe into the cell. Intracellular ROS can oxidize non-fluorescent DCFH to yield fluorescent DCF. Detection of DCF fluorescence enables assessment of intracellular ROS levels. The detailed protocol for cell staining with H2DCFDA has been incorporated into the manuscript, and the fluorescence scales have been included in Figure 5D to enhance clarity.
- The authors could have compared the findings of this study (in a separate table) with earlier reported literature (with similar works) to improve the discussion of this manuscript.
Response: The results and discussion section of the paper has been revised in accordance with relevant literature previously reported.
- The important findings and limitations of this should be highlighted in the #Conclusion section.
Response: The conclusions have been revised based on the recommendations.
References:
[1]. Lin, W.Y., et al., Probiotics and their Metabolites Reduce Oxidative Stress in Middle-Aged Mice[J]. Current Microbiology 2022. 79: 104.
[2]. Likun, L., et al., Prebiotic-Like Effects of Water Soluble Chitosan on the Intestinal Microflora in Mice[J]. International Journal of Food Engineering 2018(14): 7-8.
[3]. Zipeng, J., et al., Effect of Porcine Clostridium perfringens on Intestinal Barrier, Immunity, and Quantitative Analysis of Intestinal Bacterial Communities in Mice[J]. Frontiers in Veterinary Science 2022. 9: 881878.
Round 2
Reviewer 2 Report
Comments and Suggestions for Authors
The authors have made satisfactory revisions to the manuscript; however, the response to the question below should also be included in the main text with proper citation rather than just the response letter.
- On what basis the authors have chosen Lactobacillus, Bifidobacterium, Clostridium perfringens, and Escherichia coli as their test organisms for studying the regulatory effects of the samples on intestinal flora?
Response: Lactobacillus and Bifidobacterium are the predominant probiotics in the intestinal flora[1], while Escherichia coli represents opportunistic pathogens[2] and Clostridium perfringens represents pathogenic bacteria[3]. These four bacterial species are commonly found in the gut and can serve as indicators of gut health to a certain extent; hence, they were selected for this study.
Author Response
Editor, Foods
Reviews for: foods-2849523
Title: Effects of Pea Oligosaccharide on Probiotic Proliferation and Modulation of Inflammatory Response
Authors: Yongxia Cheng, Ruoqi Zhao, Mingwu Qiao, Yan Ma, Tiange Li, Ning Li,
Yue Shen, Xianqing Huang, Lianjun Song *
Many thanks for the reviewers’ comments. The manuscript has been updated with additional discussion and references in lines 414 to 423, addressing your suggestion regarding the fourth question.
Round 3
Reviewer 2 Report
Comments and Suggestions for Authors
The manuscript can be recommended for publication.